

# The reliability and validity of the weight-bearing lunge test in a Congenital Talipes Equinovarus population (CTEV)

Georgia Gosse[1], Emily Ward[1], Auburn McIntyre[2] and Helen A. Banwell[1,3]

[1] School of Health Sciences, University of South Australia, Adelaide, SA, Australia
[2] Physiotherapy, Allied Health Department, Women's and Children's Hospital, Adelaide, SA, Australia
[3] International Centre for Allied Health Evidence, University of South Australia, Adelaide, SA, Australia

Corresponding author
Georgia Gosse, georgiagosse07@gmail.com

## ABSTRACT

**Question**. What is the intra and inter-rater reliability and concurrent validity of the weight-bearing lunge test within a Congenital Talipes Equinovarus population?

**Design**. Test retest design for reliability and validity. The measure was taken, following preconditioning of the participants, using distance from wall, angle at distal posterior tibia using a digital inclinometer and the iPhone level function, twice by each rater. The raters included a clinician, clinician in training and a parent/carer.

**Outcome measures**. Weight bearing lunge test as a measure of ankle dorsiflexion.

**Results**. Twelve children aged 5–10 years were eligible to participate and consented, along with their parents. Intra-reliability of distance measures for all raters were good to excellent (ICC clinician 0.95, ICC training clinician 0.98 and ICC parent 0.89). Intra-rater reliability of the iPhone for all raters was good (ICCs > 0.751) and good to excellent for the inclinometer (ICC clinician 0.87, ICC training clinician 0.90). Concurrent validity between the clinician's and parents distance measure was also high with ICC of 0.899. Inter-rater reliability was excellent for distance measure (ICC = 0.948), good for the inclinometer (ICC = 0.801) and moderate for the iPhone (ICC = 0.68). Standard error of measurement ranged from 0.70–2.05, whilst the minimal detectable change ranged from 1.90–5.70.

**Conclusion**. The use of the WBLT within this CTEV population has demonstrated good to excellent reliability and validity amongst clinicians, clinicians in training and parents/carers, supporting its use as an assessment measure of dorsiflexion range of motion. There is support for parents/carers to use the WBLT at home as a monitoring assessment measure which may assist with early detection of a relapse.

**Trial registration**. University of South Australia's ethics committee (ID: 201397); Women's and Children's Hospital ethics committee (AU/1/4BD7310).

## BACKGROUND

Congenital Talipes Equinovarus (CTEV), frequently known as clubfoot, is a congenital, idiopathic abnormality affecting the lower limb in newborns (*Ansar et al., 2018*; *Symeonidis et al., 2016*). Global prevalence of CTEV is approximated at 1.2 per 1,000 livebirths, with a

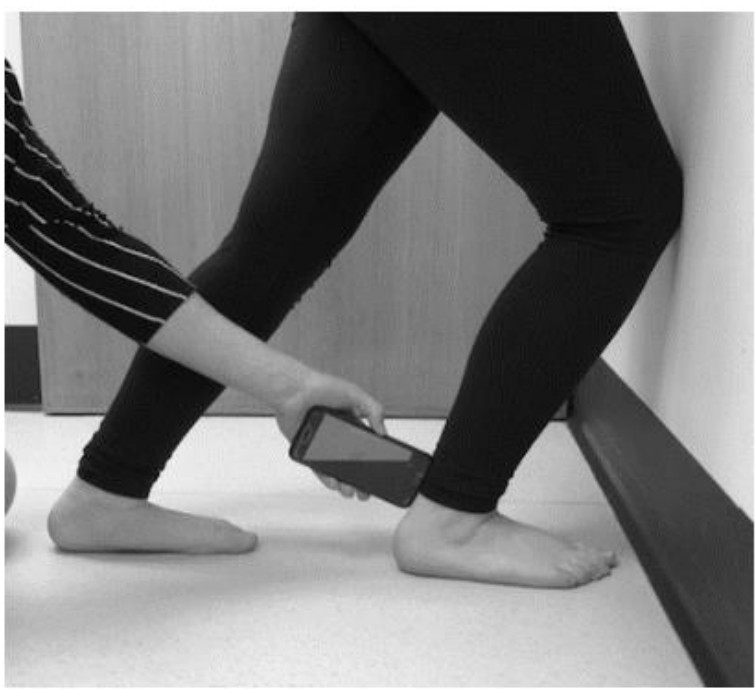

**Figure 1** Position of weight bearing lunge test with iPhone positioning and screen positioning demonstrated (author's own image).

male to female predilection of approximately 2.4:1 (*Smythe et al., 2017*). Within Australia, the Aboriginal and Torres Strait Islander population experiences a greater prevalence with 3.5 per 1,000 livebirths compared to 1.1 per 1,000 within a Caucasian population (*Ansar et al., 2018*). This condition causes the foot to be in an 'equinovarus' foot posture with adductus and cavus deformities also present (*Anand & Sala, 2008*; *Ponseti et al., 2009*; *Bergerault, Fournier & Bonnard, 2013*).

Management of CTEV via the Ponseti method includes a serial casting process of approximately six-weeks, followed by a percutaneous elongation of tendo-achilles and finally a bracing period lasting until age four (*Ponseti et al., 2009*). Unfortunately, the relapse rate remains a significant problem within this population with rates ranging from 5% to 68%, more frequently observed in those unable to comply with the bracing protocol (*Ponseti et al., 2009*; *Nogueira et al., 2013*). One study reported that at age two, the relapse rate was 30% (*Sangiorgio et al., 2017*). By the time the child was four, this was then 45% and 52% by age six (*Sangiorgio et al., 2017*).

One of the primary signs of relapse is a reduction in ankle joint range of motion (ROM) (*Ponseti et al., 2009*). The weight-bearing lunge test (WBLT), is a commonly used measure of ankle ROM (Fig. 1) (*Bennell et al., 1998*). This test has been determined as reliable within healthy adult and paediatric populations as well as some pathological groups including Charcot-Marie Tooth (*Bennell et al., 1998*; *Bennell et al., 1999*; *Rose, Burns & North, 2010*).

Monitoring of children with CTEV by health professionals decreases exponentially over time, therefore raising concern that the identification of changes in ankle joint ROM may
be delayed (*Ponseti et al., 2009*). Ideally, ankle joint ROM would be assessed regularly, more frequently than standard monitoring allows, to avoid delays in identifying those requiring further intervention (*Sangiorgio et al., 2017*). It has been reported that the use of self-management in families enhances adherence to treatment plans and provides families with greater abilities to solve problems (*Lozano & Houtrow, 2018*). This raises the consideration that parent/carers may be useful in early identification of relapses.

The WBLT can be measured in a variety of different ways, all with reported reliability and/or validity. In healthy adults, this test originally was investigated for reliability using a toe to wall measure and an angular measurement along the anterior tibia (*Bennell et al., 1998*). Another study, investigating the use of the Tiltmeter App, used the angle at the posterior tibia, measuring when the knee was both extended and flexed (*Williams, Caserta & Haines, 2013*). This study determined good to excellent reliability and validity comparing a now outdated iPhone application (the Tiltometer) with a digital inclinometer in a healthy adult population. This outcome was recently repeated using the new level function of the measure application, available within the Apple suite (Apple Inc., Cupertino, CA, USA), also with reported good to excellent reliability within a healthy adult population (*Banwell et al., 2019*). With the increase in technological advances globally, the movement of using applications in clinical settings is becoming increasingly relevant. One study found that a majority of health care providers own a smartphone with over half of those regularly using them in practice (*Franko & Tirrell, 2012*). As these tools are being used so often, it is prudent to establish their psychometric properties.

This study aims to determine the reliability and validity of two methods of measuring ankle joint ROM during the weight bearing lunge test (i.e., distance from wall and posterior angle of tibia) when conducted by a clinician, a clinician in training and a parent/caregiver.

## METHODS

This study followed a test-retest design to determine the intra and inter-rater reliabilities of the WBLT when measured by an experienced clinician, clinician in training, and the parent/carer of participants. Concurrent validity was established for the iPhone Measure app when compared to the digital inclinometer and between the experienced clinician and the parent or carer of participants. The two measures of the WBLT included distance from wall (mm) as well as posterior angle of tibia (degrees). The angle of the tibia was measured via two tools; the inclinometer within the iPhone Measure App and a digital inclinometer by the clinician and clinician in training. The parent/carer did not use the digital inclinometer due to consideration they would not have access to this tool at home. All raters were blinded to each other's measures.

### Raters

Three raters conducted each measurement. The clinician and clinician in training (AM and GG) were consistent for each participant, the third rater, a parent/carer, was unique to each participant. The clinician (AM) had thirty years clinical experience with specific involvement in paediatric orthopaedics for approximately seven years, where the WBLT is often used in practice. The clinician in training (GG) was a final year undergraduate student

and had been trained in the procedure within the previous six months. The parent/carers were not familiar with the measure but were given explanations on how to perform the test and had the opportunity to observe the raters prior to each of their measures.

The clinician and clinician in training were involved in the development of the protocol. To allow for testing and revision of protocol, the study was piloted twice (at six months and one week) prior to commencing formal study on a child with typical development.

## Participants

A sample of convenience was recruited from the Women's and Children's Hospital (Adelaide, South Australia) Physiotherapy outpatient clinic. Potential participants were identified and informed of the study by the treating clinician via a phone call or conversation when they were present for an appointment. A participant information pack was supplied where interest was indicated. Written informed consent was obtained from the parent and verbal assent gained from the child prior to commencing the measures. Participants were informed of their right to withdraw from the study via written and verbal notification.

Inclusion criteria included children aged 4–18 years born with unilateral or bilateral idiopathic CTEV, managed via the Ponseti method. The children also were required to be able to perform a WBLT without pain and have a parent/carer able to be present and conduct measures. Exclusion criteria included current pain or lower limb injury, an inability to perform the WBLT or a parent unable to measure. Reasons for being unable to measure included inability to assume a measuring posture on the floor or other physical limitations, impaired cognitive ability or previous experience in the WBLT. A sample of $n = 13$ was calculated to power the study in order to obtain 80% power, or 0.8, to detect an Intraclass Correlation Coefficient (ICC) of $\geq 0.75$ with a desired confidence interval width of 0.5 (0.5–1.0) (*Portney & Watkins, 2015*).

In the event of a child presenting with bilateral CTEV, both feet were used as separate participant data when two parents/carers were present, willing and able to measure, ensuring each parent/carer was a unique rater.

The protocol was approved by the University of South Australia Human Research Ethics Committee (approval 201397) and the Women's and Children's Hospital Research Ethics Committee (approval AU/1/4BD7310).

## Procedure

The tools used within the study included the Geo Fennel S-Digit Mini Inclinometer (digital inclinometer), (GSR Laser Tools, Perth, Australia) and the inclinometer function within the iPhone Measure application. This application is free and automatically installed on the iPhone smartphone (iOS 7 and above). Within this study, an iPhone 8 was used (Apple Inc., Cupertino, CA, USA). Prior to beginning the study, the digital inclinometer and iPhone Measure application were compared for consistency on identical, hard flat and angled surfaces across three trials. During the study the digital inclinometer was calibrated in accordance to industry requirements (Laser-Liner, UK), whilst the iPhone was calibrated to zero degrees by placing it on the long axis on the floor.

For the participants convenience, testing was conducted in conjunction to scheduled appointments. Preconditioning required participants to perform a WBLT stance for 30 s,

**Figure 2  Process of weight bearing lunge test.**

three times, to demonstrate understanding of the technique and reduce joint stiffness. A small mark was made on the back of the child's heel to indicate one centimeter superior to the posterior calcaneal tuberosity as this was the point of measurement (*Williams, Caserta & Haines, 2013*). The WBLT was performed using a modified version of methods described by previous studies and Fig. 2 shows the position in which the measure was taken (*Bennell et al., 1998*).

The measures taken included;

Clinician/Clinician in training:

1.  Distance of hallux from wall (in millimetres);
2.  Angle at back of the shin with digital inclinometer (degrees);
3.  Angle at back of the shin with iPhone Measure app inclinometer (degrees).

Parent/carer:

1.  Distance of hallux from wall (in millimetres);
2.  Angle at back of shin with iPhone measure app inclinometer (degrees).

Figure 2 describes the protocol of measures.

Unilateral CTEV participants used their affected foot. Bilateral CTEV participants with only one rater available used the foot with the higher birth Pirani score or in the case of equal scores, the child's preferred foot.

The order in which the measurements were taken were pseudo-randomised via computer programming and sealed in an envelope and labelled to corresponding participant number. For the purpose of training, the parents/carers were always the third rater. The order of the clinician and clinician in training, along with the order of measures was randomised.

The distance measure was marked on a blank piece of paper secured to the floor alongside the affected foot. If the child was unable to touch the wall with their heel on the ground, the paper was placed between the wall and the most anterior point of the knee. This resulted in a negative value. The angle measurements of the posterior leg remained the same. The measure marked on the blank piece of paper was placed in a sealed envelope until the end of the study. All distance measures were measured at the same time point at the completion of the study.

To measure the angle, the short arm of the digital inclinometer was placed flat against the posterior heel along the marked position. This was held in position, with the screen facing away from the rater for blinding until the rater stated they were pleased with the position. An independent research assistant noted the angle. The same protocol was performed with the iPhone.

Between each measure, the child was allowed to rest as needed to relieve any discomfort potentially caused by a sustained end range position and due to the child's attention span.

## DATA ANALYSIS

All data analysis was conducted using SPSS Statistics 21 software package was used (IBM Statistics, United States). Participants data were described in means (SD) and frequencies (%). The intra-rater reliability for each tool was determined using the intraclass correlation coefficients (ICC) (Model 3,1) (two-way mixed with absolute agreement), the minimal detectable change and standard error of the mean (SEM). The interrater reliability was determined using ICCs (Model 3,1) (two-way mixed with absolute agreement), SEM and the minimal detectable change. In consideration that joint stiffness may be present and impact the first measure of testing session, an apriori decision was made that the second measure of each of the raters were to be used for each of the tools. The concurrent validity of the parent/carer population was explored using ICCs (Model 2,1) (Two-way random with absolute agreement).

The minimal detectable change is the minimal amount of change that is likely not to be due to error. The SEM was used to calculate the minimal detectable change using the equation $1.96 \times \text{SEM} \times \sqrt{2}$ (*Haley & Fragala-Pinkham, 2006*). A smaller minimal detectable change is ideal as it improves confidence in difference observed, however, it does not ensure clinical relevance (*Turner et al., 2010*).

Based on an expected minimum ICC of 0.75 and a desired confidence interval (CI) width of 0.5 (i.e., the 95% CI of 0.50 to 1.00) for the intra-rater reliability analysis, it was estimated that the minimum sample size should be 13 feet.

For the reliability or validity, an ICC value of $\geq$ 0.75 with confidence interval of width 0.5 (range 0.5–1.0) was ideal. Ranges were determined, as per *Portney & Watkins, (2015)* to report ICC data: <0.5 = poor reliability, 0.5 to 0.75 = moderate reliability, 0.76 to 0.9 = good reliability, and >0.90 = excellent reliability.

All data was graphically represented on a Bland-Altmann plot. These plots provide a visual spread, illustrative of differences between methods against the mean and assists with the decision of whether the observed error is acceptable (*Portney & Watkins, 2015*). It was

**Table 1   Participant data.**

| Characteristic | Mean (±SD) | Range |
|---|---|---|
| Age (years) | 7.00 (±1.80) | 5–10 |
| Weight (kg) | 22.90 (±7.60) | 15–39 |
| Height (cm) | 121.90 (±14.60) | 102–148 |
| Shin length (cm) | 28.20 (±4.90) | 21–35 |
| Foot length (cm) | 16/60 (±2.80) | 14–22 |
| Pirani score (from birth) | 5.00 (±1.03) | 3–6 |

used to assess the degree of agreement between the two tools in all positions, by both raters, across the two timepoints.

# RESULTS

## Participant characteristics

Twelve participants and their parents/carers met eligibility criteria with both parent and child consenting to being involved in the study. Participants characteristics were recorded (Table 1). Additionally, the carer filled out a purpose-built questionnaire (Additional file 8) to determine the child's CTEV experience. Seven out of the twelve participants (58.3%) had bilateral CTEV. A slight gender bias existed with 66.7% being males (8:4), in keeping with expected gender prevalence of CTEV.

## Study findings

Measures were taken on thirteen feet (Table 2). A negative recording on the knee to wall measure (i.e., unable to touch the wall) was recorded for five (42.7%) measures. Two hundred and eight measures were recorded during the study.

The concurrent validity between the iPhone and digital inclinometer on flat and angled surface (15 degrees) was determined prior to the study. The validity was excellent, indicated by an ICC of 0.99 (95% confidence interval −0.58 to 1.58).

The intra-rater reliability between measures for the distance measure was excellent (ICC = 0.96–0.99), very good for the digital inclinometer (ICC = 0.85–0.90) and good for the iPhone measure app (ICC = 0.75–0.90) (Table 3). Inter-rater reliability between the clinician and clinician in training was excellent using the distance measure (ICC = 0.95), good when using the inclinometer (ICC = 0.80) and moderate for the iPhone measure application (ICC = 0.68) (Table 3).

The standard error of measurement (SEM) and minimal detectable change was determined for the intra-reliability of each of the measures (Table 3). The minimal detectable change ranged from 1.90–5.70 with the clinician in training's measures, using the digital inclinometer, having the lowest minimal detectable change.

Concurrent validity between the clinician and parent/carer was good (ICC = 0.90) for distance as displayed by the Bland-Altmann plot below. The iPhone tool provided moderate validity between the clinician and parent/carer (ICC = 0.62).

The Bland-Altmann plot (Fig. 3) shows the agreement between the clinician and parents/carers distance. All data points, except for one outlier, were between the limits

Peerj

**Table 2  Raw measurements.** Table two shows measure two taken by each rater for each participant.

**TABLE OF RAW RESULTS**

| Foot # | Distance score (mm) | | | Angle at back of tibia (inclinometer) (degrees) | | Angle at back of tibia (iPhone compass app) (degrees) | | |
|---|---|---|---|---|---|---|---|---|
| | Clinician | Clinician in training | Parent/Carer | Clinician | Clinician in training | Clinician | Clinician in training | Parent/Carer |
| 1 | −40 | −71 | −22 | 19.4 | 19.4 | 16 | 16 | 17 |
| 2 | −30 | −29 | −28 | 29.6 | 28.8 | 11 | 25 | 25 |
| 3 | −10 | 0 | 5 | 26.2 | 22.8 | 23 | 21 | 24 |
| 4 | 10 | 0 | 0 | 27.5 | 27.5 | 27 | 28 | 28 |
| 5 | 19 | 12 | 21 | 23.9 | 20.7 | 24 | 20 | 24 |
| 6 | −11 | −18 | −3 | 23.8 | 22.4 | 26 | 22 | 23 |
| 7 | 6 | 0 | 6 | 19.8 | 24 | 25 | 25 | 20 |
| 8 | 0 | 0 | 15 | 26 | 23 | 27 | 24 | 28 |
| 9 | −1 | −16 | −12 | 20.6 | 19.1 | 17 | 18 | 18 |
| 10 | 32 | 40 | 66 | 27.1 | 28.2 | 26 | 28 | 33 |
| 11 | 17 | 23 | 35 | 17.1 | 24.3 | 16 | 19 | 24 |
| 12 | 29 | 27 | 50 | 27.2 | 27.8 | 25 | 26 | 29 |
| 13 | 16 | 22 | 19 | 25.6 | 26 | 24 | 26 | 24 |

**Table 3  Study results.** Table of raw measurements.

**INTRA-RATER RELIABILITIES**

| | Rater | Mean (SD) | ICC | 95% CI | SEM | MDC |
|---|---|---|---|---|---|---|
| Digital inclinometer (degrees) | Clinician | −1.50 (±2.30) | 0.87 | 0.52, 0.96 | 0.83 | 2.30 |
| | Clinician in training | 0.90 (2.20) | 0.90 | 0.68, 0.97 | 0.70 | 1.93 |
| iPhone (degrees) | Clinician | −0.50 (4.10) | 0.75 | 0.16, 0.92 | 2.05 | 5.68 |
| | Clinician in training | 0.30 (2.60) | 0.90 | 0.68, 0.97 | 0.82 | 2.28 |
| | PC | −1.80 (2.40) | 0.90 | 0.49, 0.97 | 0.76 | 2.10 |
| Distance (mm) | Clinician | −2.20 (10.00) | 0.96 | 0.86, 0.99 | 2.00 | 5.54 |
| | Clinician in training | −2.00 (7.10) | 0.98 | 0.96, 0.99 | 1.00 | 2.78 |
| | PC | 0.43 (7.80) | 0.97 | 0.88, 0.99 | 1.35 | 3.74 |

**INTER-RATER RELIABILITIES**

| | Raters | Mean (SD) | ICC | 95% CI | | |
|---|---|---|---|---|---|---|
| Digital inclinometer (degrees) | Clinician/clinician in training | −0.01 (2.90) | 0.80 | 0.32–0.94 | | |
| iPhone (degrees) | Clinician/clinician in training | −0.90 (4.60) | 0.68 | 0.06–0.90 | | |
| Distance (mm) | Clinician/clinician in training | 3.60 (11.10) | 0.95 | 0.84–0.98 | | |

**CRITERION VALIDITY**

| | Raters | Mean (SD) | ICC | 95% CI | | |
|---|---|---|---|---|---|---|
| iPhone (degrees) | Clinician/PC | −2.3 (4.90) | 0.62 | −0.11, 0.88 | | |
| Distance (mm) | Clinician/PC | −8.8 (12.80) | 0.89 | 0.58, 0.97 | | |

**Notes.**
Abbreviations: SD, standard deviation; ICC, intraclass correlation coefficient; CI, confidence interval; SEM, standard error of measurement; MDC, minimal detectable change; PC, parent/carer.

of agreement. This demonstrates the consistency and therefore concurrent validity of the measures.

## DISCUSSION

This study is the first to explore the reliability of the WBLT within a CTEV population. The WBLT is used by clinicians to assess ankle joint ROM and has been deemed reliable within pathological paediatric populations, such as Charcot-Marie Toot (*Rose, Burns & North, 2010*) calcaneal apophysitis (*James et al., 2015*) and idiopathic toe walking (*Williams et al., 2013*). The current study followed the protocol of these previous studies, which is an adapted version of the original WBLT by Bennell and Talbot (*Bennell et al., 1998*). This study has determined that identifying a change in ankle joint ROM using distance of toes from wall, and inclinometer has good to excellent intra and inter-rater reliability and iPhone measure has good intra-reliability. The measures can be used with credence by parents/carers to identify change in ankle ROM, potentially indicating early CTEV relapse. As stated earlier, a reduction in ankle joint ROM is one of the primary signs of relapse and early detection of change leads to earlier intervention (*Ponseti et al., 2009*; *Sangiorgio et al., 2017*).

The literature reports the relapse involved with CTEV continues to be high. Children with CTEV are reviewed by health professionals less frequently as they grow older; at a time when their risk for relapse continues (*Ponseti et al., 2009*). Having parents/carers able to identify early changes in ankle joint ROM improves monitoring abilities, detecting joint

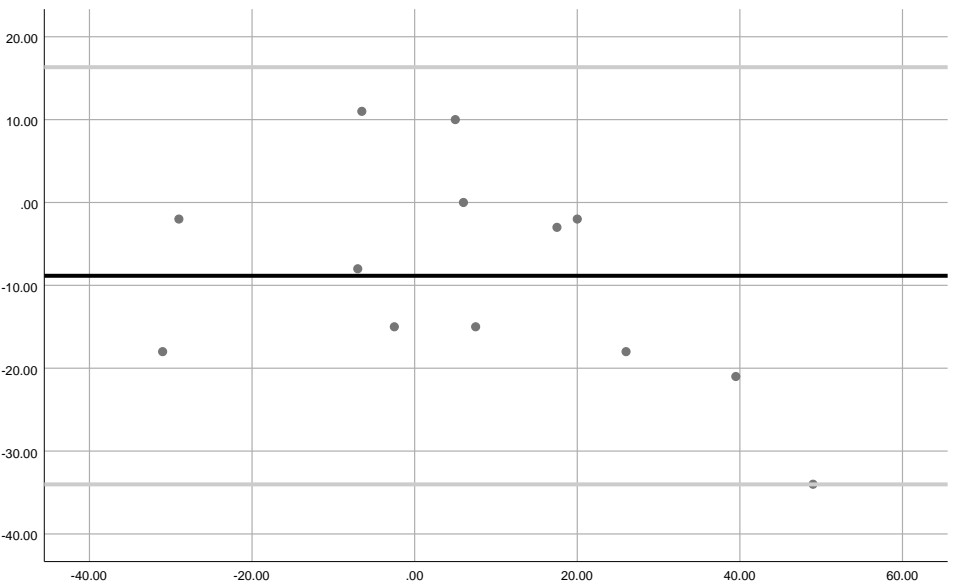

**Figure 3** Bland-Altmann demonstrating agreement between clinicians and parents/carers distance measure (concurrent validity).

changes and relapses sooner for better outcomes. Our sample was a population presenting to a metropolitan hospital were but given the ease of this measure, it can be used anywhere. This is particularly significant in the Aboriginal and Torres Strait Islander community where there is a much higher prevalence of CTEV. Given 11.9% of Aboriginal and Torres Strait Islander people live in areas classified as very remote, and due to inherent difficulties in receiving adequate health-care in remote areas, a heavier reliance on self-monitoring is required (*Australian Bureau of Statistics, 2018*). The use of simple tools like the distance or measure application can allow people to identify concerns with their own health and seek more timely and appropriate intervention.

The distance measure proved to be most reliable from the WBLT measure options reviewed, potentially due to ease of application. However, this study determined the WBLT within a CTEV population can be measured by a variety of people, in a variety of ways, with confidence. It is noted the low minimal detectable change results across all measures suggest a small change in measure cannot be attributed to an error in measurement and further boosts confidence that measurers are observing true change. These results are in keeping with previous investigations of the reliability and validity WBLT in adult, paediatric and pathological populations (*Banwell et al., 2019*).

These outcomes should be considered against a number of limitations. Firstly, due to the CTEV presentation, the children measured had feet with a soft heel and rounded lateral border (Fig. 4). This potentially increased the difficulty of obtaining consistent measures.

The inquisitive nature of the children along with the repetitive nature of three measuring tools, lead to frequent movement, with children attempting to change body position to gain a better view of what was occurring. This occasionally meant there was some movement

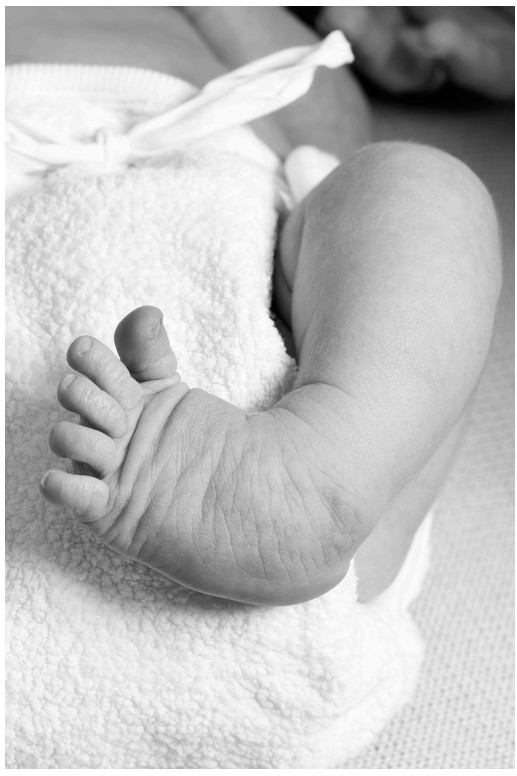

**Figure 4** Example of clubfoot with rounded lateral border. Figure source: iStock.

of the foot, requiring realignment. It is also important to mention also that this study only measured ankle dorsiflexion. A relapse of CTEV could, potentially, occur in multiple planes due to the nature of the condition. It is important that this is deliberated when considered for application. This study only measured the reliability of an iPhone with regards to phone type. The results are therefore most relevant to Apple users. Although the distance measure can be used by all and is most reliable, there is potential to assess this measure using different technologies. Future studies are required for the long term follow up of the use of the WBLT by carers as a self-monitoring tool. This should be followed in relation to reported relapse identification. Particularly in remote areas to determine the efficiency of the tool.

Converse to much of the literature the clinician in training and parent/carer were observed to have moderately smaller SEMs to the clinician (*Mueller et al., 1989*). This may reflect care and concentration of the novice users and suggests future studies should consider using more than one representative for each user group involved.

Future studies should involve the development and testing of a WBL protocol for use at home by parents/carers in relation to the sensitivity and specificity of the measure. This

protocol could involve a prospective long-term investigation prior to determining if the WBLT measure alone is competent in detecting a CTEV relapse in the home setting.

## CONCLUSION

The WBLT within a paediatric CTEV population has good to excellent reliability when used by either a clinician, clinician in training or parent/carer, for distance from the wall, or the angle of the posterior lower leg when using an inclinometer or iPhone (intra-reliability only). Good concurrent validity is also demonstrated for the distance measure. The results of this study are encouraging as a tool for increasing self-monitoring of this condition and potential earlier detection of relapses. This will be particularly useful in remote areas with limited health-care services. Ankle dorsiflexion is, however, just one of the signs of relapse and it would be prudent for clinicians to consider other signs and symptoms prior to diagnosis. Future studies should aim to develop a protocol for this measure at home with parents and test the effectiveness of relapse prediction and associated outcomes.

## ACKNOWLEDGEMENTS

Thank you to the University of South Australia and their library staff for their assistance with the search of literature. Thank you also to the Women's and Children's Hospital in Adelaide for the use of their resources.

### Funding
The authors received no funding for this work.

### Competing Interests
The authors declare they have no competing interests.

### Author Contributions
- Georgia Gosse conceived and designed the experiments, performed the experiments, analyzed the data, prepared figures and/or tables, authored or reviewed drafts of the paper, and approved the final draft.
- Emily Ward and Helen Banwell conceived and designed the experiments, authored or reviewed drafts of the paper, and approved the final draft.
- Auburn McIntyre conceived and designed the experiments, performed the experiments, authored or reviewed drafts of the paper, and approved the final draft.

### Human Ethics
The following information was supplied relating to ethical approvals (i.e., approving body and any reference numbers):

University of South Australia ethics committee (Approval number ID: 201397) and the Women's and Children's Hospital ethics committee (AU/1/4BD7310)

## Data Availability

The raw measurements are available as a Supplementary File.

## Supplemental Information

Supplemental information for this article can be found online at http://dx.doi.org/10.7717/peerj.10253#supplemental-information.

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
