# Peer review of "The reliability and validity of the weight-bearing lunge test in a Congenital Talipes Equinovarus population (CTEV)"

_PeerJ, doi:10.7717/peerj.10253_

## Round 0.1 · original submission · Major Revisions

Dear authors,

In light of the reviewers' comments, I think your work has scientific merit to be considered for publication in PeerJ. However, there are some issues which you must address in a revised version of your manuscript.

Best regards,
Dr Palazón-Bru (academic editor for PeerJ)

·

Basic reporting

Thank you for the opportunity to review this paper. Overall, I feel this paper is well written and provides new knowledge in the space of home monitoring of CTEV. In particular, providing data which is objective. This paper is well written and clear throughout.


The abstract is well written and the background provides a good context. The structure is clear and the figures and tables are appropriate. The results are relevant to the hypothesis.

Line 100 – the authors refer to a ‘six-week casting process. Consider adding – ‘approximately’ as some take longer.

Line 103 – states ‘in those who do not comply’. Consider reframing to ‘unable to comply’. There is some anectodal thought in this space that compliance may be related to an uncompliant foot rather than an incompliant patient/ family.

Line 107 – states that ‘The primary sign of relapse is a reduction in ankle range of motion’. As this is a deformity which occurs in multiple planes, relapse can also occur in multiple planes. Consider revising to ‘One of the primary signs of relapse is a reduction in ankle range of motion’.



Line 140 – consider changing WBL to WBLT for consistency or define WBL in addition to WBLT.

Experimental design

Line 112 – This is a good paragraph which highlights the need for regular monitoring. Consider adding a sentence around early identification of relapse to minimise the intervention required. This could link nicely to the final sentence around ‘useful in early identification of relapses’.

Sample size was justified well.

Methods – consider stating if participants were blinded to each other’s measures.

Line 172 – Inclusion criteria – Please state if idiopathic cases were only included, or if atypical, those associated with syndromes included.

Line 257 – good justification to use the second measure.

Overall the study was designed well with good rigorous measures such as blinding of the measures until the investigator was happy with the placement. There is sufficient detail to replicate this project.

Validity of the findings

Table 2 presents reliability and validity data. I am wondering if the Mean (SD) column could please be revised for greater clarity. I am unable to ascertain what this data specifically represents.
I would be interested to know what ranges / averages / median (appropriate measure) were found in this cohort? My query relates to whether, if there was wide variability in the ranges we could be more confident that these tools could measure a wide range of DF scores reliably. If, however, the range of DF was small, we could only be confident in the reliability within that range. It may be possible that the data provided covers this, apologies if this was the data in Table 2 which I misinterpreted.

Conclusions are well stated and within the scope of the project. Overall data is robust and statistically sound.

·

Basic reporting

Basic reporting: The style and clarity is of high standard. The referencing depth and breadth could be improved upon with regards the statistical tools/rationales as per the PDF comments provided.
The introduction is succinct and effective. The discussion is also well tethered to the rest of the paper and sets out useful future work that is needed.
The figures seem easy to comprehend and of use. The raw data is provided and is easy to digest especially considering the small sample size required for such studies.

Experimental design

Experimental design: The selected methods fit the aims of the paper well. The procedure used could be expanded upon as per PDF comments to enhance further. The statistical methods used seem sound and logical but enhanced referencing/justification could be added. Good, honest mention of the challenge of getting inquisitive children to do something repeatedly whilst trying to blind them from the results.

Validity of the findings

Validity of the findings: The impact of the findings should be significant especially in such cases and with remote, vulnerable communities. If you can provide info regarding the samples community backgrounds that would enhance the impact and the external validity of the claimed benefits. There needs to be re-wording around the link to clinical relevance as per the PDF comments made. The limitations are open and honest as well as considerate of the wider picture that is not entirely captured by this single direction measurement.

Additional comments

Your paper is well written and of great interest to such a large population. The methods chosen to investigate seem appropriate and reflective of current telehealth and remote technology use movements/developments.
The paper is of use and with the aforementioned changes should be of great use to a wide audience. The topic of accurate measurement and enabling patients/families to take responsibility for such methods is of use outside of this pathology group too. The paper flows well and is easy to read and digest.

---

## Round 0.2 · accepted · Accept

All the reviewers' comments have been correctly addressed.

·

Basic reporting

No comment

Experimental design

No comment

Validity of the findings

No comment

Additional comments

The revisions in this paper have improved the overall work. I congratulate the authors on this work.